# Arsenic in Mining Areas: Environmental Contamination Routes

**DOI:** 10.3390/ijerph20054291

**Published:** 2023-02-28

**Authors:** Márcia Cristina da Silva Faria, Rodrigo de Carvalho Hott, Maicon Junior dos Santos, Mayra Soares Santos, Thainá Gusmão Andrade, Cleide Aparecida Bomfeti, Bruno Alves Rocha, Fernando Barbosa, Jairo Lisboa Rodrigues

**Affiliations:** 1Instituto de Ciência, Engenharia e Tecnologia (ICET), Universidade Federal dos Vales do Jequitinhonha e Mucuri (UFVJM), Teófilo Otoni 39803-371, MG, Brazil; 2Analytical and System Toxicology Laboratory, Department of Clinical Analyses, Toxicology and Food Sciences, School of Pharmaceutical Sciences of Ribeirao Preto, University of Sao Paulo, Ribeirao Preto 14040-903, SP, Brazil

**Keywords:** toxic metal, particulate matter, risk, environmental monitoring

## Abstract

The emission and accumulation of toxic elements such as arsenic in various environmental compartments have become increasingly frequent primarily due to anthropogenic actions such as those observed in agricultural, industrial, and mining activities. An example of environmental arsenic contamination in Brazil exists in the city of Paracatu, MG, due to the operation of a gold mine. The aim of this work is to evaluate the routes and effects of arsenic contamination in environmental compartments (air, water, and soil) and environmental organisms (fish and vegetables) from mining regions as well as the trophic transfer of the element for a risk assessment of the population. In this study, high levels of arsenic were found in the waters of the Rico stream ranging from 4.05 µg/L during the summer season to 72.4 µg/L during the winter season. Moreover, the highest As concentration was 1.668 mg kg^−1^ in soil samples, which are influenced by seasonal variation and by proximity to the gold mine. Inorganic and organic arsenic species were found above the allowed limit in biological samples, indicating the transfer of arsenic found in the environment and demonstrating a great risk to the population exposed to this area. This study demonstrates the importance of environmental monitoring to diagnose contamination and encourage the search for new interventions and risk assessments for the population.

## 1. Introduction

Arsenic (As) is a grayish-colored crystalline metalloid with an atomic number of 33 and an atomic mass of 74.9 g·mol^−1^ [1]. It is listed first among substances toxic to human health by the Agency for Toxic Substances and Disease Registry (ATSDR) from the Department of Health of the United States and thus recognized as an important carcinogen [2,3].

This element is widely distributed in the biosphere and can be found in the atmosphere, water, soils, and organisms [4,5,6,7]. Despite being considered a trace element, its concentration in the environment has become increasingly greater due to natural phenomena such as weathering and volcanic eruptions and anthropogenic actions such as the release of domestic and industrial effluents, pesticides, and, above all, mining activity [8]. In the environment, the toxicity of a metal depends on the oxidation state and chemical structure of the metallic species. Arsenic exists in nature in a variety of chemical forms, including organic species, such as monomethylarsonic acid (MMA), dimethylarsinic acid (DMA), arsenobetaine (AsB), and arsenocholine (AsC), and inorganic species, which are the most toxic and are formed by arsenate (As^5+^) and arsenite (As^3+^). Notably, the inorganic compounds of As are about 100 times more toxic than the partially methylated organic forms (MMA and DMA), since the increasing order of toxicity of the As compounds is as follows: organic compounds of As^5+^ < compounds organic of As^3+^ < compounds inorganic of As^5+^ < compounds inorganic of As^3+^ [9]. A caveat must be made regarding DNA damage, where the trivalent methylated arsenic species, that is, MMA (III) and DMA (III), are considered more toxic than inorganic arsenic because they are more efficient in causing the breaking of DNA [10].

Most As in the environment is associated with rocks and minerals; it is present in more than 200 minerals, primarily as arsenate and arsenic sulfides and to a lesser degree as arsenites, oxides, and elemental As [11]. As has a high affinity with some elements, such as sulfur and iron, making sulfate minerals, for example, generate low mobility to other environmental compartments [12]. Arsenopyrite is the most common sulfide mineral and is generally associated with gold (Au) mining. Its oxidation, which is dependent on several factors, including the amount of arsenopyrite, distribution of the size of the ore, temperature, and time of the exposure of the material to water and oxygen [13], produces arsenic oxyacids that facilitate its release into the soil and watercourse [14].

Its excretion occurs through metabolism in the liver, where As^5+^ is reduced by the enzyme arsenate reductase to As^3+^, which is methylated to methyl arsonate (MMA^5+^) and, sequentially, to dimethyl arsenic acid (DMA^5+^) by the As-methyl transferase enzyme. This conversion from As-i to As-o allows its excretion in urine [15,16]. The amount of As^3+^, As^5+^, MMA, and DMA in urine may vary according to exposure conditions, the route of administration, and dose. This urinary excretion ends up becoming part of domestic sewage, thus representing one more factor increasing As in rivers and streams that receive domestic effluents without treatment [17].

Mining activities are the main anthropic source of As release to the environment. Gold mine tailings and effluents generally contain high concentrations of arsenic and are a cause for concern as potential sources of environmental contamination [18]. In addition to solid and liquid tailings, mining is also responsible for releasing As into the atmosphere along with particulate matter that is suspended in the air during processing in gold mines. As is removed from the atmosphere by dry deposition or rain. Some airborne particles can condense into dust particles that can be inhaled or ingested [19].

The impact of particulate material (PM) containing As on humans is mainly related to health and generally associated with the inhalation of particles less than 10 μm in diameter. Breathable particles (<2.5 μm in diameter) can reach the pulmonary alveoli [20]. Although there is no national legislation that determines the amount of As in the air, the ATSDR [1] establishes values from 20 to 100 ng·m^3^ in remote and urban areas, respectively. Values below those recommended by ATSDR that are chronically inhaled and associated with other sources of As contamination, such as via food and water, have already been related to the increased incidence of cancer [21].

One way to assess the impact of environmental contamination by As is through the analysis of animal and plant species that are exposed to this metal. As has a great capacity to generate biomagnification at different trophic levels of the food chain, and its presence in the environment is quickly detected in organisms exposed to that environment, with such detection being a strong indicator of risk to humans [22]. In addition to detecting the presence of As, from a food security perspective, it is also essential to determine the species and their quantities in the food consumed by humans, allowing a more accurate assessment of health risk, as different species of As exhibit different physical–chemical properties and toxicities [10,23].

Gold mining is one of the main factors responsible for the increase in As in many ecosystems on the planet through effluents, soil leaching, and dust particles released during the mining process, affecting plants, animals, and humans. As a result, populations, especially those close to industrial and geological sources, are exposed to higher levels of As in various ways, such as via food, drinking water, soil, and ambient air, causing serious health impacts [24].

The occurrence, distribution, origin, and mobility of As have received significant attention in recent years. The knowledge of the processes that govern the accumulation of As in the environment as well as the possible toxic effects on populations in affected areas is necessary for a better understanding. This study is carried out in the city of Paracatu, MG, which is known for gold mining that has existed since the 18th century, occurring mainly in the vicinity of the Rico stream, an important tributary of the Paracatu River Basin and whose source is located in a mining area of gold cutting across the entire urban area along its path [11]. Urban proximity to mining areas, whose soil is rich in arsenopyrite, can generate environmental contamination with arsenic released by human action and consequently put the population at risk. In this scenario, given the great influence that mining areas exert on urban areas and knowing the health risks of exposure to As, it is important to determine its concentration in water, soil, and particulate matter as well as its availability in animal species and vegetables [11].

The aim of this work is to evaluate the routes and effects of arsenic contamination in environmental compartments (air, water, and soil) and in environmental organisms (fish and vegetables) from a mining region and to evaluate the trophic transfer of the element for a risk assessment of the population.

## 2. Materials and Methods

### 2.1. Study Location and Collection Points

The study was carried out in Paracatu, a city in the State of Minas Gerais with approximately 85,000 inhabitants and the largest open-pit gold mine in Brazil, “Morro do Ouro”. It is located 2 km from the urban center of the municipality [25], at latitude 17°13′21″ S and longitude 46°52′31″ W, near the Rico stream. This mine is located in the sub-basin of the Paracatu River, which belongs to the São Francisco River Basin [11].

Environmental samples, such as water from the Rico stream, soil from its banks, and particulate matter from the atmosphere, as well as biological samples such as fish and plants were collected to assess levels of contamination by As. Physical–chemical parameters were also determined: pH, turbidity, and concentrations of Mg^2+^ and Ca^2+^ ions. Sample collection took place during the winter of 2017 (dry season) and summer of 2018 (rainy season). Three collections of water and soil were made at five different points along the Rico stream (collection points 1, 2, 3, 4 and 5; Figure 1). The first point was close to the source within the mining area, three points were in the urban region, and the final point was outside of the urban area. Sample collections were performed according to the procedure adopted by the Environmental Company of the State of São Paulo [26].

A total of 1000 mL of water was collected in triplicate in polypropylene metal-free flasks. The soil samples were collected in triplicate in a polypropylene container containing 500 g of the samples by using a 20 cm drill. The particulate material (PM) samples were taken in two regions of the city known as “Amoreiras” and “Chapadinha”. The Amoreiras region is close to the mining region, while the Chapadinha region is located at the southern end of the city, farther from the mining region (Figure 1). Samples of corn, cassava, and fish were collected in the Rico stream area during the winter of 2017. The samples were stored in metal-free plastic bags and packed in refrigerated coolers for less than 24 h.

### 2.2. Materials and Chemicals

All reagents used in the experiments were of analytical grade and used without further purification. The reagents used were weighed on a Shimadzu analytical scale, model AUY220, to prepare the solutions. The solutions were prepared using ultrapure water (resistivity 18.0 MΩ cm) obtained by the Thermo Scientific Barnstead™ Nanopure™ system (Waltham, MA, USA).

The reagents Na_2_HAsO_4_·7H_2_O (CAS: 10048-95-0) and NaAsO_2_ (CAS: 7784-46-5) were provided by Sigma-Aldrich (St. Louis, MO, USA), and the reagents NaOH (CAS: 1310-73-2), HNO_3_ (CAS: 7697-37-2), CH_3_COOH (CAS: 64-19-7), and HCl (CAS: 7647-01-0) were purchased from Vetec (Brazil). Rhodium (Ra) (CAS: 7440-14-4) and multi-elementary solution were obtained from PerkinElmer (Waltham, MA, USA).

### 2.3. Analysis of Arsenic from Environmental Samples

#### 2.3.1. Quantification of Arsenic in the Samples

All experiments were performed in triplicate in a clean room (class 1000). Calibration curves in the range of 1 to 50 μg L^−1^ (As) were prepared with analytical standards, and the limit of detection found was 0.009 μg L^−1^. All samples analyzed were diluted in 2% (*v*/*v*) nitric acid solution (HNO_3_) using the protocol established by EPA 200.8 [27].

The concentrations of arsenic were determined with an inductively coupled plasma mass spectrometry (ICP–MS) NexION 300D da PerkinElmer (EUA) with a Meinhard nebulizer and quartz cyclonic spray chamber with continuous nebulization. The operating conditions were as follows: nebulizer gas flow rate, 0.95 L min^−1^; auxiliary gas flow, 1.2 L min^−1^; plasma gas flow, 15 L min^−1^; lens voltage, 7.25 V; ICP RF power, 1300 W; CeO/Ce = 0.011; and Ba^++^/Ba^+^ = 0.016.

#### 2.3.2. Soil Samples

The soil samples were dried at 60 °C for 48 h and sieved (20 µm). Afterward, 1.0 g was subjected to acid extraction with 10 mL of distilled HNO_3_ assisted by a CEM microwave digester, model MARS 6, according to method 3051A of the United States Environmental Protection Agency [28]. After digestion, the aliquots of the solution were analyzed by ICP-MS. The data were analyzed by ANOVA with 95% confidence intervals and Tukey’s test with a significance level of α = 0.05.

#### 2.3.3. Water Samples

Aliquots of the water samples were diluted in 2% HNO_3_ solution and analyzed by Method 3051a from the United States Environmental Protection Agency [28]. After dilution, aliquots of the solutions were analyzed by ICP-MS. The data were analyzed by ANOVA with 95% confidence intervals and Tukey’s test with a significance level of α = 0.05.

#### 2.3.4. Particulate Material Samples

Particulate matter up to 2.5 µm (PM 2.5) and total suspended particulate matter (PTS) were collected by Pall Life Sciences glass fiber membrane filters with 24 h cycles using a large volume sampler for the determination of the concentration of respirable particles (up to 2.5 µm) in ambient air (AGV MP2.5) with a separation head and volumetric flow controller (CVV) and a large volume sampler (AGV) for the determination the concentration of PTS in ambient air, both manufactured by Energica. The sampling method was carried out based on the rules for sample collection (EPA 2006) and in accordance with the National Environment Council (CONAMA) [29].

To determine the As levels in the PM present in the filters, adapted method IO-3.1 from the American Environmental Protection Agency was used (US EPA) [30]. The filters were dried (24 h) and weighed before and after PM collection. The weight difference corresponded to the total mass collected. Then, the particles were subjected to acid extraction with 10 mL of distilled HNO_3_ assisted by a CEM microwave digester, model MARS 6, according to US EPA method 3051A [31].

The accuracy of the method was proven by the preparation and analysis of a matrix peak using a blank filter and a sample filter added with standard As ICP solution and certified reference material (CRM) NIST 1648a. A 2.5 cm × 20.3 cm blank glass fiber filter strip was spiked with 1 mL of 1 mg L^−1^ As ICP standard solution. Another blank strip was spiked with 0.03 g of CRM NIST 1648a (urban particulate matter), and a sampled 2.5 cm × 20.3 cm fiberglass filter strip was spiked with 1 mL of 1 mg L^−1^ As ICP standard solution. The fortified strips were applied and followed the same procedure described above. After digestion, the aliquots of the solution were analyzed by ICP-MS. The data were analyzed by ANOVA with 95% confidence intervals and Tukey’s test with significance level of α = 0.05.

### 2.4. Determination of Hydrogen Potential (pH), Water Hardness, and Turbidity

In the water samples collected in the Rico stream, determinations of hydrogen potential and calcium carbonate concentration (hardness) were performed to assess the retention and/or release of metals from sediments into the water [32].

For the determination of the pH, a pH meter manufactured by Digimed model DM-2P was used. The determination of water hardness was carried out using the Aquamerk determination kit (Merk^®^, Darmstadt, Germany). To determine the turbidity of the water samples, a Hanna instruments turbidimeter model HI93703C was used [32].

### 2.5. Quantification of As Species in Maize, Cassava, and Fish

Corn, a cereal, and cassava, a tuberous root, are foods widely grown and consumed in Brazil. Due to their high absorption of As, they can act as a vehicle of this metal in the food chain, generating bioaccumulation and serving as a good bioindicator of environmental contamination. Another important indicator used to measure the contamination of As in aquatic ecosystems is fish, which can accumulate this metal through water, food, and sediments, leading to the biomagnification of metals [33]. In this work, we used samples of the axial muscle of the fish Traíra (*Hoplias malabaricus*) and Dourado (*Salminus maxillosus*), the grains of corn (*Zea mays* L.), and the tuberous part of the cassava (*Manihot esculenta*) without the husk.

Briefly, the corn, cassava, and fish muscle samples were washed, crushed, lyophilized, and sieved. After this preparation, 100 mg of each sample was weighed in triplicate so that they could be pre-digested (48 h) using 2.0 mL of sub-distilled HNO_3_. Then, the samples were heated (90 °C) for 4 h in a microwave oven (model MARS 6), and the volume was made up to 50 mL with ultrapure water [34]. After digestion, the samples were cooled to room temperature, filtered with a 0.20 µm cellulose filter, and injected directly into the LC-ICP-MS. The samples were loaded with a syringe into a 100 µL sample loop. All chromatographic separations were performed with a PRP-X100 anion exchange column (150 mm × 4.6 mm, 5 µm) (Hamilton, Reno, NV, USA) at 25 °C under isocratic conditions. The mobile phase, with a flow rate of 1 mL min^−1^, was composed of 10 mmol L^−1^ HPO_4_^2−^/H_2_PO_4_^−^ (pH 8.5) (98% (*v*/*v*)) and methanol (2% (*v*/*v*)). Analytical calibration standards for As species (As^3+^, As^5+^, DMA, and MMA) were prepared in the range of 1.0–20.0 µg L^−1^ by serial dilutions. The data were analyzed by ANOVA with 95% confidence intervals and Tukey’s test with a significance level of α = 0.05.

### 2.6. Allium Cepa Test

The *Allium cepa* test is an important bioassay for monitoring the potential synergistic effects of a mixture of pollutants, including toxic metals, allowing the determination of the cytotoxicity, genotoxicity, and mutagenicity of As present in water [35]. The present study was carried out according to method described in a previous study, with modifications [35].

To perform the test, 100 seeds of *Allium cepa* were used for each water sample from the Rico stream and were spread over a Petri dish containing filter paper moistened with 6 mL of the sample. As a negative control, ultrapure water (18 MΩ cm) was used, and CuSO4 (0.006 mg·mL^−1^) was used as a positive control. The experiment was maintained in a germination chamber (Thermo Scientific, Waltham, MA, USA) at 25 °C under constant light for 5 days. After this period, the growing roots were measured with the aid of a caliper ruler.

The rate of root inhibition was assessed by the average length of growing seeds (ALS) [35]. To obtain the ALS, the arithmetic mean was calculated according to the root sizes. To determine the inhibition index, Equation (1) was applied:Inhibition rate = (Negative control ALS) − (ALS at the analyzed point) × 100(1)

To determine the mitotic index, genotoxicity, and mutagenicity of samples from the Rico stream, the roots were cut and fixed with Carnoy’s solution for 6 h and later replaced by 1 mL of 70% ethanol for a further 6 h. After this period, the roots were washed with ultrapure water and then subjected to hydrolysis with a 1 M HCl solution in a water bath at 60 °C for 9 min and stained with Schiff’s reagent for 2 h in the dark. Then, the root was cut, placed on a slide, and a drop of 2% acetic carmine solution was added. After 8 min, it was covered with coverslips and carefully kneaded. To determine the IM, Equation (2) was used:MI = MCN/TCN × 100(2)
where MCN corresponds to the number of mitotic cells and CTN means the total number of cells analyzed. All cells in mitosis were quantified and analyzed. Chromosomal aberrations caused by changes in the structure or the total number of chromosomes were identified to determine genotoxicity. An increase in the number of micronuclei was used to determine the occurrence of mutagenicity.

## 3. Results and Discussion

### 3.1. Physicochemical Parameters and Arsenic Concentration in Surface Water and Soil on the Banks of the Rico Stream

The physical–chemical parameters of the surface waters of the Rico stream in the municipality of Paracatu are listed in Table 1.

The samples collected at the five points of the Rico stream in the dry and rainy seasons had pH values close to neutrality, ranging from 6.0 to 7.1, and were therefore in accordance with Brazilian legislation [36].

Hardness represents the measure of water’s ability to precipitate soap due to the presence of calcium and magnesium ions [38] and is therefore calculated from the concentrations of these ions [37]. Samples P1 and P2, collected in the rainy season, were classified as moderately hard due to the presence of a large amount of Ca^2+^ and Mg^2+^ in the water close to the spring accentuated by weathering in the rainy season. Following the course of the stream, as it enters the city, a drop in hardness occurs due to the natural sedimentation of Ca^2+^ and Mg^2+^. This hardness increases again after leaving the city due to the deposition of untreated domestic sewage, which became more evident in P5 collected in the dry period, where there was a lower volume of water and the same volume of sewage.

Turbidity is a property that assesses how difficult it is for water to transmit light due to the presence of suspended materials such as clay, organic matter, plankton, and other microscopic organisms. The main sources of suspended materials are particles originating from weathering processes that occur in rocks and soils in the hydrographic basin, being directly related to mining activities, deforestation, and the discharge of domestic effluents [39]. The waters of the studied region showed turbidity ranging from 0.71 to 8.3 UNT with no major differences between samples collected in the dry season and rainy season. The turbidity values proved to be low and within the requirements of legislation.

To analyze the As values in the surface waters of the Rico stream, we considered the CONAMA 357 resolution [36] and information from the World Health Organization [40], which classifies the studied stream waters as class 2 and determines the maximum concentration of As to be 10 µg L^−1^ for water to be used for human supply (after conventional treatment), primary contact recreation, the protection of aquatic communities, irrigation, aquaculture, and fishing activities.

We found that the only surface water sample collected in the summer (rainy season) that showed a high concentration of As was point 5 with a value 300% above the limit (Figure 2A). With regard to the samples collected in the winter (dry season), all five points showed increased values for As, which is a matter of concern in terms of the consumption and use of water for irrigation. It is important to mention that collection site 1 was closest to the mining area. The high values of As found in the water of site 5, both in the dry and rainy seasons, can instead be explained by the natural presence of As in the soils of the studied region and by the residential sewage discharged into the stream.

Seasonal behavior was observed since all the samples collected during the summer showed lower levels than those collected in the winter. Although regional rains during the summer cause the leaching of arsenic-rich soils [44], they also end up diluting the arsenic naturally found in these waters due to the increased volume of water in the rivers and streams. It is also necessary to consider that the mineral form of arsenopyrite, which is very common in the mining region of Paracatu, can be oxidized, releasing As into the soil and watercourse due to the influence of some factors such as particle size distribution, low pH values, heat, and exposure to water and oxygen [12]. These factors are less influential in the rainy season due to the increase in rainwater. 

The average annual humidity in the city of Paracatu is around 72%, with the months from June to August being the driest and the months from December to January being the most humid. The monthly averages of the rainfall index between the years 1973 and 2018 are shown in Figure 3.

The concentration of As in the soils was above that established by CONAMA resolution number 420 of 2009, which establishes an acceptable value to be below 15 mg kg^−1^ and considers intervention values in agricultural, residential, and industrial regions to be above 35, 55, and 150 mg kg^−1^, respectively [44].

According to the results of the analysis of As in the soil along the Rico stream (Figure 2B), all the sampled points were well above the minimum levels required by legislation, with values exceeding more than one hundred times the established limit. The highest values were found in the locations closest to the gold mine, with values collected during the rainy season of 1.502 and 1.545 mg kg^−1^ at points 1 and 2, respectively. For the dry season, the concentration found was 1.668 mg kg^−1^ at point 1 and 1.257 mg kg^−1^ at point 2. The values found are in agreement with those reported by Rezende et al. (2015), who detected concentrations of arsenic in soils in the city of Paracatu ranging from 32 to 2.980 mg kg^−1^ [11].

The Rico stream can present concentrations 190 times higher than that stipulated by environmental legislation and 744 times higher than the average natural concentrations of rivers and streams in the region, with high levels of As associated with natural sources of the Paracatu region and the exploration of gold [11]. According to the Minas Gerais Water Management Institute [45], concentrations of As above the level considered safe by CONAMA [43] can cause several adverse effects on the biological community. According to CETESB [26], the values found classify the Paracatu area as contaminated with potential risks, direct or indirect, to human health, and the area is considered a scenario of wide exposure.

### 3.2. Determination of As Concentration in Particulate Matter (PM) of Atmospheric Air in the City of Paracatu, MG

The filters containing the total suspended particles (TSP) in the atmosphere from the samplers in the municipality of Paracatu were analyzed quantitatively for arsenic (As). The geographic locations of the sampling stations are shown in Figure 1.

The As recovery values for the filter blank with a standard solution and with the addition of NIST 1648a were 96% and 91%, respectively. For the sampled strip enriched with the standard As solution, the As recovery was 98%. All recovery values obtained are within the limits proposed by the US-EPA (blank filter: 80 to 120%; sampled filter: 75 to 125%) [30].

The results (Figure 2C) show that the filters in the Mulberry region had the highest concentrations of As, demonstrating a greater risk for the population residing near the gold mine. It is also observed that in the summer, there is an increase in the amount of particulate matter in the city of Paracatu as it is a period with the occurrence of higher temperatures and a higher incidence of gusty winds. It is noted that wind gusts can occasionally increase the resuspension of soil particles and dust dispersion.

In rural areas, the average concentrations of As are considered to range from 1 to 4 ng m^3^, while in urban areas, they range from 5 to 7 ng m^3^ (measurements carried out in the United Kingdom). In the United States, As concentrations are estimated to range from 1 to 5 ng m^3^ in remote/rural areas and from 20 to 100 ng m^3^ in urban areas [2]. However, in methodologies for assessing environmental risks to human health due to exposure to As in the atmosphere via inhalation, with respect to carcinogenic effects, the average reference levels of As concentration for acceptable risk range from 0.2 ng m^3^ to 0.6 ng m^3^ for risks of 1:1,000,000 (10^−6^) and from 2 ng m^3^ to 6 ng m^3^ for risks of 1:100,000 (10^−5^) [31]. 

With respect to non-carcinogenic effects, the California Environmental Protection Agency CalEPA established the reference maximum level of concentration of As in the atmosphere by chronic inhalation as 30 ng m^3^. This limit represents a concentration below which toxic, non-carcinogenic effects are not expected to appear in the human population [46].

Although all the As values in the PM of this study were below the recommended limits, accumulation generated by the intake of As present in water, food, and PM can reach high levels in humans, making it a cause for concern. Mortality and cancer increase in regions where the atmospheric As is greater than 2 ng m^3^ [21]. 

### 3.3. Quantification of As Species (As^3+^, As^5+^, MMA, and DMA) in Samples of Corn, Cassava, and Fish Exposed to Water and Soil on the Banks of the Rico Stream

Figure 4 presents the results for the speciation of arsenic in the samples of fish (traíra and dourado), corn, and manioc. All the samples showed a total value of As (*∑*As) above the limits established by RDC 42 [47].

The samples of traíra (fish) showed the highest concentrations of As with a sum between the species of As of 2.794 mg kg^−1^, which is above the maximum limit of 1.0 mg kg^−1^. The As^3+^ species was the most prevalent with 2.233 mg kg^−1^ followed by As^5+^ and DMA with 0.381 and 0.180 mg kg^−1^, respectively. The traíra belongs to the family Erythrinidae, whose main characteristics are those of carnivorous and predatory fish feeding on other fish, frogs, and insects, preferably inhabiting lentic environments and having nocturnal habits. They are located next to mud bottoms or in rocky locations and have a high resistance to places with little oxygen [48]. These characteristics explain their higher concentrations of As^3+^, which is normally present in sediments and in places with little oxygen. Golden dourado fish, on the other hand, had a total As concentration of 2.64 mg kg^−1^, which is also above the 1.0 mg kg^−1^ limit. It presented 2.63 mg kg^−1^ of DMA and 0.01 mg kg^−1^ of MMA. The dourado is also a species of carnivorous fish that feeds on smaller fish, but it also consumes aquatic plants and is usually found in running waters.

The differences in the fish species with respect to the kinetic behavior (accumulation and elimination) of As were observed for other fish species as well as there being a correlation established between the fish family and the speciation patterns of As [49]. In a recent experiment, the accumulation of DMA in goldfish was mainly attributed to biotransformation rather than trophic transfer. Within the fish organism, biotransformation and detoxification mechanisms have been reported, such as the reduction of inorganic arsenic received through diet or water followed by methylation to less toxic organic forms [50]. Inorganic arsenic is methylated via oxidative methylation, forming first MMA and then DMA. An alternative methylation scheme in which MMA and DMA are produced using a common As^3+^-triglutathione complex has also been proposed. In both cases, the sequential formation of DMA can explain its positive correlation [51].

Regarding the results obtained for the cassava samples, total As values (0.758 mg kg^−1^) were also well above the legal limit of 0.1 mg kg^−1^. As^5+^ was the most prevalent species at 0.441 mg kg^−1^ followed by DMA and As^3+^ at 0.247 and 0.07 mg kg^−1^, respectively.

Corn also showed values above the 0.2 mg kg^−1^ limit with 0.405 mg kg^−1^ total As, which represented the sum of As^5+^, DMA, and MMA species at values of 0.31, 0.075, and 0.02 mg kg^−1^, respectively. When some plant species such as corn and cassava are grown in environments rich in As, it is common for the bioaccumulation process to be exhibited [52]. As^5+^ was shown to be the most bioavailable for both cassava and maize, which represents the greatest risk for human consumption since the inorganic species of As are the most toxic and remain the longest in the body, carrying out more connections with biological structures. Such results corroborate the high As values found in water and soil in this region (Figure 2).

### 3.4. Allium Cepa Test

#### 3.4.1. Analysis of Toxicity of Rico Stream Waters

The seeds submerged in the different samples of the Rico stream showed a lower root growth rate than those obtained in the negative control. The results of the cytotoxicity test expressed in Table 2 demonstrate that the roots of *Allium cepa* were susceptible to toxic compounds present in the samples, causing an inhibition that prevented the normal growth of the roots. The sites close to mining showed significant inhibition values as can be seen at points 1, 2, and 3. Although location 5 was furthest from the mining area, it presented the highest rate of inhibition. This point also showed the highest concentration of As in soil and water samples.

As can interfere with root growth as a result of the inhibition of the electron transport chain, replacing phosphate with arsenate and thus causing a reduction in energy production during cellular respiration [53,54].

#### 3.4.2. Mitotic Index

A mitotic index significantly lower than that of the negative control may represent changes originating from the action of chemical compounds on the growth and development of the exposed organisms, while an MI greater than that of the negative control as a result of the growth of cell division could be harmful to cells, resulting in disordered cell proliferation [55]. All the samples had an MI lower than the negative control (Table 3), proving the existence of toxic substances in the analyzed water. Studies have shown that a reduction in cell activity may be due to changes in the duration of the mitotic cycle [56], which can contribute to cytotoxicity.

#### 3.4.3. Genotoxicity Analysis

Chromosomal aberrations evaluated for genotoxicity analysis are characterized by changes in chromosomal structures or in the total number of chromosomes considered at different stages of cell division, which can occur spontaneously or by exposure to contaminants. Genotoxicity studies were carried out by analyzing the chromosomal aberrations (CAs) present in the slides prepared for this study. The changes considered for this study were chromosomal breaks, C-metaphase, losses, bridges, sticky chromosomes, and spindle abnormalities (Figure 5). Approximately 5000 cells were evaluated per sample.

All the samples analyzed showed an increase in chromosomal aberrations present in meristematic cells in cell division compared with the negative control, indicating genotoxicity. As noted in Table 4, points 1, 2, and 3 near the gold mine as well as the more distant points 4 and 5 presented significant chromosomal aberrations in relation to the control, proving the As genotoxic effect in As-rich stream waters. Some studies [57,58] suggested that viscosity may be due to the degradation or depolymerization of chromosomal DNA. Viscosity was also attributed to the entanglement of interchromosomal chromatin fibers. Viscosity is a common sign of a toxic influence on chromosomes and is probably an irreversible effect. It can represent a great risk for human populations that consume water with a high concentration of arsenic, with observed genotoxic effects.

#### 3.4.4. Mutagenicity Analysis

An analysis of mutagenic potential was performed based on the frequencies of meristematic cells with micronuclei. The micronucleus arises from the development of certain chromosomal aberrations, such as chromosomal breaks and losses, by the development of a new membrane that involves a piece of chromatin that failed to pass to the poles during the anaphase of cell division. The analysis was performed for all phases of the cell cycle (interphase, prophase, metaphase, anaphase, and telophase); 5000 cells counted for each sample were studied, and the results are shown in Figure 6.

In this study, all the samples showed a statistically greater occurrence of micronuclei in relation to the negative control (*p* < 0.05), indicating mutagenic potential. This increase was significantly greater in samples that contained a greater amount of As, as observed in other studies. According to a work by Faita et al., 2013, exposure to As promoted an increase in the DNA fragmentation rate and the frequency of micronuclei in addition to a drop in the MI, with these responses being dependent on the concentration and time of exposure to the pollutant [58].

A genotoxic and mutagenic evaluation of effluents containing metals was observed in the work of Matsumoto et al., 2006 [59], where samples of water from the Bagres stream (Brazil), which receives effluents from tanneries, caused significant induction of chromosomal aberrations and micronuclei. Studies carried out using the *Allium cepa* test on the Sava river (Croatia), which is impacted by urban, industrial, and agricultural effluents, revealed the inhibition of root growth, variations in the mitotic index, high frequencies of chromosomal aberration, and micronuclei in the cells analyzed [55].

## 4. Conclusions

In this study, high levels of arsenic were found in soil samples and in the waters of the Rico stream, which were influenced by seasonal variation and by proximity to the gold mine, indicating that although the region’s soil is rich in arsenic minerals such as arsenopyrite, processing has a great influence on the dispersion of this semimetal in the environment.

In addition, there was an increase in As in areas with the discharge of domestic effluent, indicating the presence of As in the bodies of residents of the region. As values above the limit were found in biological samples (fish, corn, and cassava), indicating the trophic transfer of this element and characterizing ecological damage that affects species in this area and puts human health at risk.

This study demonstrated that there is a need to reduce the As exposure of the population living close to this mining area and to use efficient methods to promote environmental remediation of the affected area.

## Figures and Tables

**Figure 1 ijerph-20-04291-f001:**
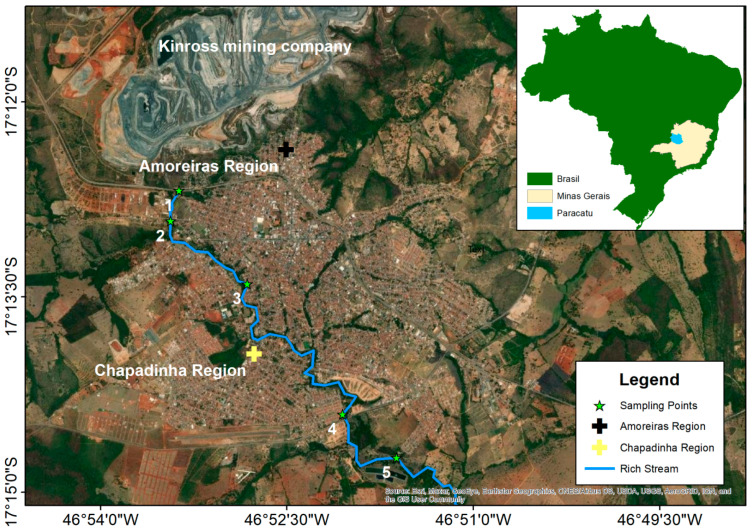
Map of the city of Paracatu, MG, with the points of water and soil sample collection. Software used is ArcGis version 10.8.

**Figure 2 ijerph-20-04291-f002:**
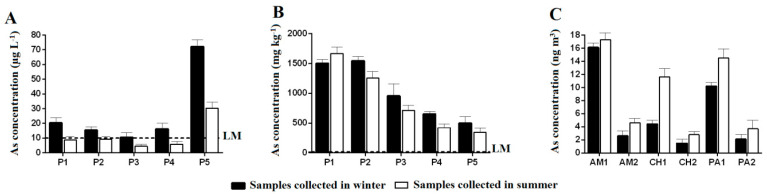
Results of the analysis of arsenic in water (**A**) and soil (**B**) at points P1, P2, P3, P4, and P5 of the Rico stream and atmosphere (**C**). Particulate matter up to 2.5 µm (PM 2.5) and particulate matter in total suspension (PTS) collected in the regions of Amoreiras (AM), Chapadinha (CH), and Paracatu (PA). Analyses were carried out in the winter of 2017 and summer of 2018. The limit (LM) is <10 µg L^−1^ for surface water [36,41] and <15 mg kg^−1^ for soil [42]; >35 mg kg^−1^ requires agricultural intervention [43]. α < 0.05.

**Figure 3 ijerph-20-04291-f003:**
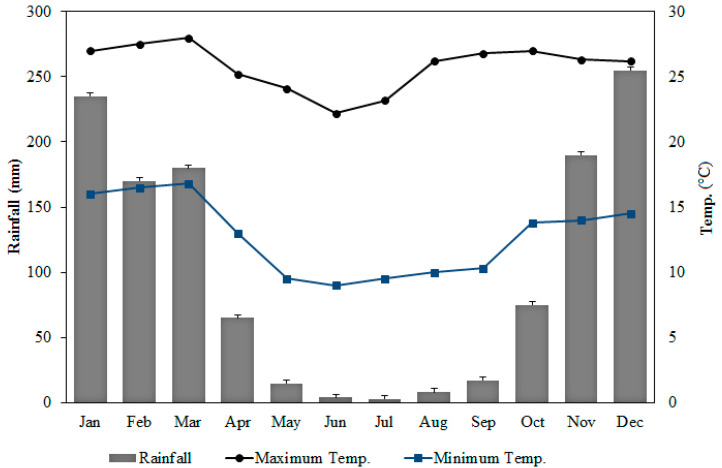
Average rainfall index (mm) and average maximum and minimum temperatures (°C) over the months from 1973 to 2018. Measurements obtained by the Brazilian National Institute of Meteorology.

**Figure 4 ijerph-20-04291-f004:**
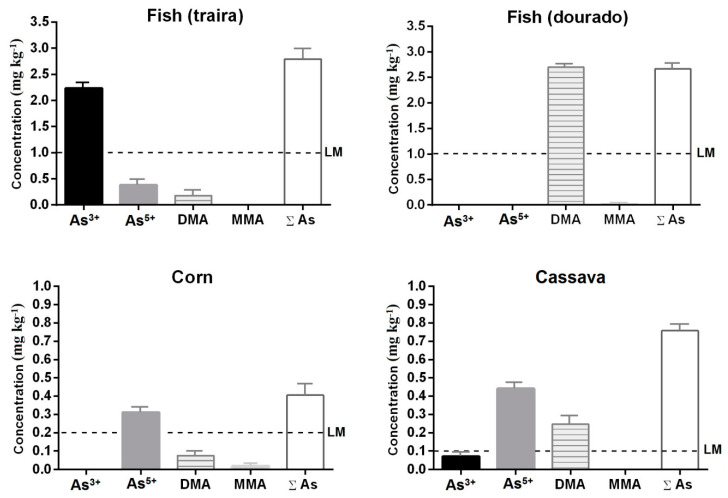
Quantification of As species (As^3+^, As^5+^, MMA, and DMA) in samples of corn, cassava, and fish exposed to water and soil on the banks of the Rico stream. ∑As = sum of As species; LM = maximum limit of As (LM fish 1.0 mg kg^−1^, LM corn 0.2 mg kg^−1^, and LM cassava 0.1 mg kg^−1^) [47].

**Figure 5 ijerph-20-04291-f005:**
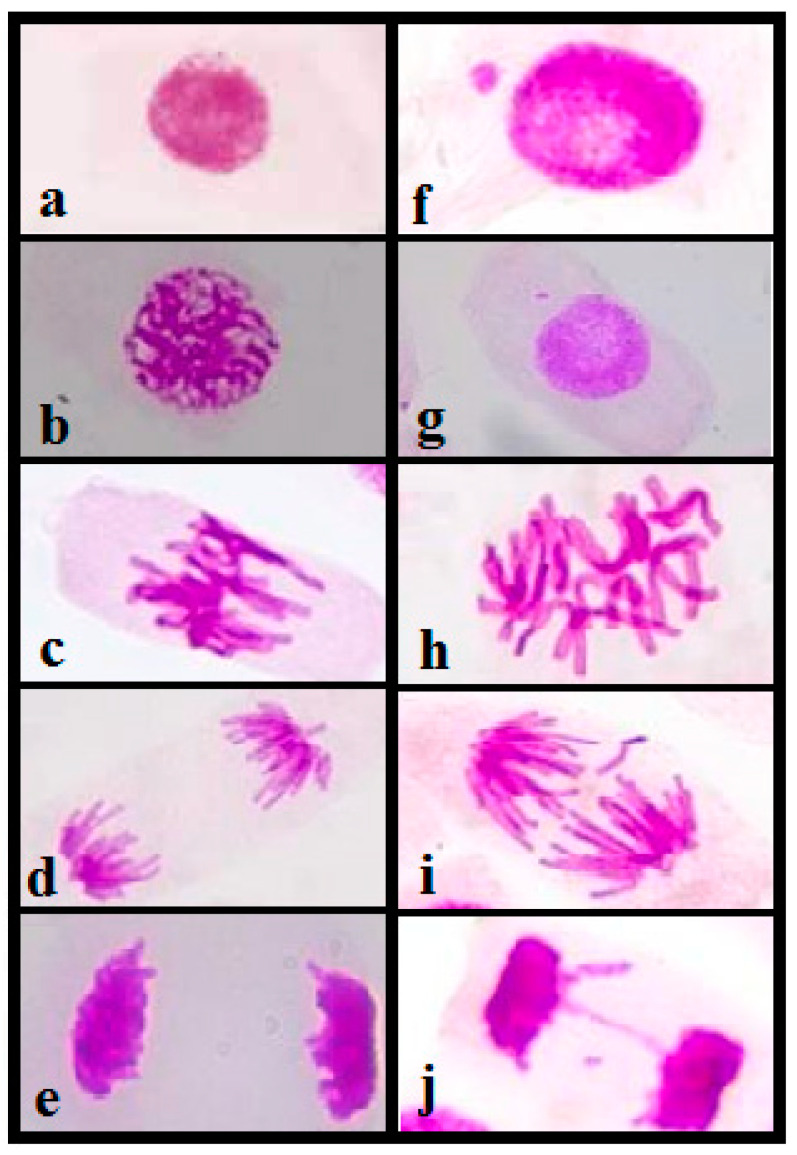
Microscopic image of the mitosis subphases of meristematic cells from the root of *Allium cepa*: (**a**) interphase; (**b**) prophase; (**c**) metaphase; (**d**) anaphase; (**e**) telophase and chromosomal aberrations (CAs); (**f**) micronucleus interphase; (**g**) micronucleus prophase; (**h**) C-metaphase; (**i**) anaphase with chromosomal loss; and (**j**) telophase with bridge and break.

**Figure 6 ijerph-20-04291-f006:**
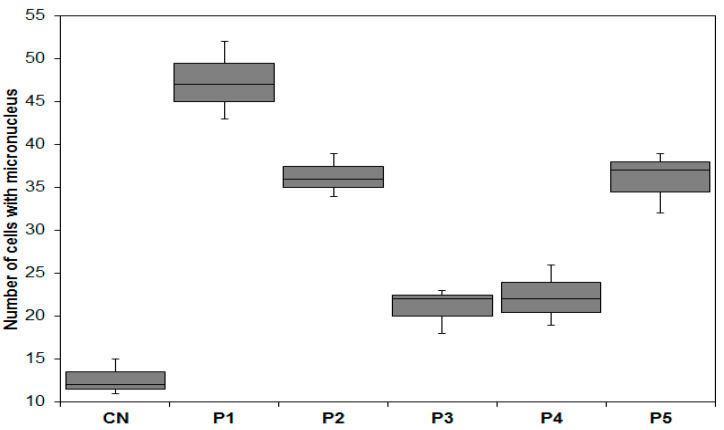
Occurrence of micronuclei in *Allium cepa* meristematic cells exposed to samples collected in the Rico stream (P1, P2, P3, P4, and P5) and negative control of ultrapure water (NC) for 24 h. Tukey test with significance level of q = 0.05.

**Table 1 ijerph-20-04291-t001:** The physical–chemical parameters (pH, hardness, and turbidity) of the surface waters of the Rico stream at the five selected points. The collections were carried out during the winter of 2017 (dry season) and the summer of 2018 (rainy season). Reference values: pH (6–9), turbidity <40 UNT, hardness (soft: <50 mg L^−1^ CaCO_3_; moderate: between 50 mg L^−1^ and 150 mg L^−1^ CaCO_3_; hard: between 150 mg L^−1^ and 300 mg L^−1^ CaCO_3_; and very hard: >300 mg L^−1^ CaCO3 (potability limit 500 mg L^−1^ CaCO_3_)) [36,37].

Analysis References		Collection Points				
		P1	P2	P3	P4	P5
pH Summer	6.0–9.0	6.0	6.5	6.9	7.0	7.0
pH Winter		6.4	6.8	7.1	7.1	7.1
Ca^2+^ and MgCa^2+^ (mg L^−1^) Summer	-	94.0	88.0	29.0	40.0	47.0
CaCa^2+^ and MgCa^2+^ (mg L^−1^) Winter		49.0	41.0	19.0	35.0	48.0
Turbidity (UNT) Summer	<40	1.7	0.87	2.1	1.0	3.0
Turbidity (UNT) Winter		3.5	8.3	1.6	0.71	2.1

**Table 2 ijerph-20-04291-t002:** Average growth (cm) and rate of inhibition of *Allium cepa* roots grown in a water sample from the Rico stream and collected in winter of 2017 at the five points of the study. Ultrapure water (18 MΩ cm) was used for negative control (NC). ALR is the average length of growing seeds. Different letters indicate significantly different values between treatments (*p* < 0.05).

Samples	P1	P2	P3	P4	P5	NC
ALR (cm)	0.25 ± 0.3 ^a^	0.27 ± 0.2 ^a^	0.18 ± 0.2 ^b^	0.37 ± 0.2 ^c^	0.12 ± 0.1 ^d^	0.42 ± 0.2 ^e^
Inhibition rate	41.00%	37.00%	55.30%	11.31%	66.22%	-

**Table 3 ijerph-20-04291-t003:** Total mean mitotic indices (MIs) obtained from meristematic cells (in mitosis) of *Allium cepa* root cultivated with water from the Rico stream and collected in winter of 2017 at the five points of the study. Ultrapure water (18 MΩ cm) was used for negative control (NC). Shown are mitotic indices of 5.000 cells per group. Different letters indicate significantly different values between treatments (*p* < 0.05).

Samples	P1	P2	P3	P4	P5	NC
MI	2.66% ^a^	2.03% ^b^	1.75% ^c^	1.93% ^c^	2.20% ^b^	5.96% ^d^

**Table 4 ijerph-20-04291-t004:** Chromosomal aberrations (CAs). Ultrapure water was used for negative control (NC). Shows the number of cells analyzed (NCA). Different letters indicate significantly different values between treatments (*p* < 0.05).

Samples	NCA	Bridges	Losses	Breaks	Sticky Chromosomes	C-Metaphase	Spindle Anomalies	CA
NC	5000	2	2	19	0	4	0	27 ^a^
Water P1	5000	9	3	28	20	24	2	86 ^b^
Water P2	5000	7	1	22	20	17	1	68 ^c^
Water P3	5000	5	0	17	18	12	0	52 ^d^
Water P4	5000	4	3	16	5	12	0	40 ^e^
Water P5	5000	7	4	23	22	21	3	80 ^b^

## Data Availability

The datasets generated during the current study are available from the authors upon reasonable request.

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
