# Peer review of "Arsenic in Mining Areas: Environmental Contamination Routes"

_ijerph, 2023, doi:10.3390/ijerph20054291_

Round 1

Reviewer 1 Report

See attached file

Author Response

Dear editor and reviewer:

Reviewer 2 Report

Manuscript deals with the "environmental contamination of arsenic by mining activities".

As is the 20th abundant component on the Earth’s crust. However, As is a toxic metalloid and is remarked as a considerable global groundwater contaminant, affecting certain rivers and deltas in East and South Asia and in South American countries. Based on the Agency for Toxic Substances and Disease Registry list 2017, As is amongst the most hazardous materials that could be poisonous to humans. Approximately 200 million people in around 70 countries have been exposed to this metalloid. As enters the environments via natural sources, such as rocks, As-enriched minerals, forest fires, volcanoes and anthropogenic sources (e.g., mining, herbicides, phosphate fertilisers, smelting, industrial processes, coal combustion and timber preservatives).

1. Page 1, "..high levels of arsenic are found in soil samples and in the waters of the Rico stream,.." Mention the values.

2. As listed below, there are several studies about Arsenic contamination and ecotoxicology in the Paracatu area. Therefore, may you please clarify the novelty of your study?

https://doi.org/10.1016/j.ecoenv.2021.112869

https://www.cetem.gov.br/antigo/images/programas/paracatu/Ecotoxicological-assessment-of-arsenic-contaminated-soil-and-freshwater-from-Paracatu-Minas-Gerais-Brazil.pdf

https://assets.researchsquare.com/files/rs-481921/v1/847866a3-a782-425b-981e-8e2184c8420f.pdf?c=1631882431

https://kinross.com.br/duvidas-frequentes/paracatu-arsenic-study-results-and-references/

3. Please improve the quality of Fig. 2.

Author Response

Dear Editor and Reviewer 2:

Round 2

Reviewer 1 Report

The authors did a great job of writing the article and correcting the comments of the reviewers. The article is suitable for publication in its present form.